# The Dynamic Change in Aromatic Compounds and Their Relationship with *CsAAAT* Genes during the Post-Harvest Process of Oolong Tea

**DOI:** 10.3390/metabo13070868

**Published:** 2023-07-20

**Authors:** Ziwei Zhou, Qingyang Wu, Hongting Rao, Liewei Cai, Shizhong Zheng, Yun Sun

**Affiliations:** 1College of Life Science, Ningde Normal University, Ningde 352100, China; t2133@ndnu.edu.cn (Z.Z.); 1150311011@fafu.edu.cn (H.R.); t1935@ndnu.edu.cn (L.C.); t1127@ndnu.edu.cn (S.Z.); 2Key Laboratory of Tea Science in Fujian Province, College of Horticulture, Fujian Agriculture and Forestry University, Fuzhou 350002, China; 1210311021@fafu.edu.cn

**Keywords:** oolong tea, manufacturing process, volatile phenylpropanoids/benzenoids, *CsAAAT* gene

## Abstract

Formed by L-phenylalanine (L-phe) ammonia under the action of aromatic amino acid aminotransferases (AAATs), volatile benzenoids (VBs) and volatile phenylpropanoids (VPs) are essential aromatic components in oolong tea (*Camellia sinensis*). However, the key VB/VP components responsible for the aromatic quality of oolong tea need to be revealed, and the formation mechanism of VBs/VPs based on AAAT branches during the post-harvest process of oolong tea remains unclear. Therefore, in this study, raw oolong tea and manufacturing samples were used as the test materials, and targeted metabolomics combined with transcriptome analysis was also conducted. The results showed that thirteen types of VBs/VPs were identified, including nine types of VPs and four types of VBs. Based on the OAV calculation, in raw oolong tea, 2-hydroxy benzoic acid methyl ester and phenylethyl alcohol were identified as key components of the aromatic quality of oolong tea. As for the results from the selection of related genes, firstly, a total of sixteen candidate *CsAAAT* genes were selected and divided into two sub-families (*CsAAAT1* and *CsAAAT2*); then, six key *CsAAAT* genes closely related to VB/VP formation were screened. The upregulation of the expression level of *CsAAAT2*-type genes may respond to light stress during solar-withering as well as the mechanical force of turnover. This study can help to understand the formation mechanism of aromatic compounds during oolong tea processing and provide a theoretical reference for future research on the formation of naturally floral and fruity aromas in oolong tea.

## 1. Introduction

The six major types of tea in China are distinguished by their different manufacturing methods, among which oolong tea has the most complicated procedure. Due to its unique history and culture, abundant germplasm resources, and exquisite processing technology, oolong tea is highly fragrant with a mellow taste, and it is deeply favored by consumers [1,2]. Aroma is one of the key factors determining the quality of tea [3]. The aromatic quality of oolong tea is affected by tea varieties, growth environments, cultivation conditions, and, more importantly, the multiple stress treatments of the fresh leaves during the post-harvest procedure [4,5,6]. Most volatile organic compounds (VOCs) form and accumulate during the post-harvest procedure of oolong tea [7]. The endogenous enzyme biochemical reaction caused by turnover and the physical–chemical reaction caused by fermentation as well as drying are two of the main reasons for VOC formation and accumulation during the post-harvest process of oolong tea [8,9]. It is noteworthy that turnover is the core method for forming the characteristic floral and fruity aroma quality of oolong tea [10].

The formation of oolong tea’s aroma relies on four metabolic pathways: the terpene pathway, the fatty acid pathway, the benzene ring/phenylpropane pathway, and the carotenoid pathway [11,12]. Among these, phenylpropanoids/benzenoids, the second most common volatile components in the plant kingdom, play a vital role in determining the aroma of tea.

L-phe is an important product in the shikimic acid metabolism pathway. There are three main branches that form VBs/VPs, taking L-phe as a precursor under enzymatic reactions. Firstly, under the reaction of phenylalanine lyase (PAL), the intermediate product trans-cinnamic acid forms, followed by VBs such as benzaldehyde, benzyl alcohol, benzoic acid, and salicylic acid, which are generated through enzymatic and non-enzymatic reactions [13]. Secondly, phenylacetaldehyde synthase (PAAS) directly reacts with L-phe to form VPs, including phenylacetaldehyde, phenylethanol, and phenylethyl ester [14]. Thirdly, L-phe can also participate in the formation of VBs/VPs after forming phenylpyruvic acid (PA) under the reaction of aromatic amino acid transaminase (AAAT), which has been confirmed in watermelon, rose, camellia, and some other horticultural crops [15,16,17]. Wang et al. (2019) reported that PA was obtained from L-phe under the catalysis of *CsAAAT* using isotope tracer technology [18]. In addition, the *CsAAAT1* and *CsAAAT2* genes, with a high catalytic efficiency for L-phe and tyrosine, were screened for the first time. Furthermore, in terms of the quality of oolong tea products, an increasing amount of research points to the contribution of VBs/VPs in oolong tea’s flavor quality. Previous studies found that VBs/VPs, including indole [19], benzaldehyde and benzyl alcohol [20,21], phenylethanol and phenylacetaldehyde [22,23], and hexyl benzoate and cis-3-hexenyl benzoate [24,25], were formed during the post-harvest process of oolong tea, which would eventually contribute to the aromatic quality of oolong tea (Figure 1).

However, the specific VB/VP components that contribute to the aromatic quality of oolong tea remain unclear. Compared to previous studies on PAL and PAAS branches, the AAAT branch has received less attention. Even less is known about the transcription mechanism of *CsAAAT* genes and its influence on the dynamic changes in VBs/VPs during the post-harvest process of oolong tea. Therefore, in this study, oolong tea manufacturing process samples were utilized as materials. The metabolic profiles of the VBs/VPs were constructed, and then based on a tea genome database (CSS) and transcriptome database, the transcripts of key *CsAAAT* genes related to VB/VP changes were screened. The purpose of this study was to investigate the formation mechanism of VBs/VPs, in order to provide a theoretical basis for the formation of aroma components during the post-harvest process of oolong tea.

## 2. Materials and Methods

### 2.1. Plant Materials and Treatments

Tea leaves were collected from *Camellia sinensis* cv. Huangdan, a major tea variety in the oolong tea production area, grown at the educational practice base (26 04′ N, 119 14′ E) of Fujian Agriculture and Forestry University (Fuzhou, China), from 2 to 3 p.m. on 30 September 2019, under sunny conditions. Processing technology for oolong tea originating from southern Fujian was applied to the post-harvest treatment of fresh leaves. The standard for plucking was “one bud and three leaves”, which means one bud and three leaves on the same tea plant branch. The treatment involved the usual methods and stimulation that are common in the oolong tea manufacturing process, which are typical for the process, industrially [34]. The fresh leaves (F) were solar-withered under sunlight (26 °C, 120,000 Lx) for 0.5 h. Then, a portion of solar-withered leaves (SW, 500 g) was turned over once per hour for 5 min each time, with three times in total. After that, a sample of tea leaves was taken and marked as T3. The remaining half of the SW was used as the control group for spreading without turnover, being sampled and marked as CK3.

Subsequently, according to the manufacturing methods of oolong tea, samples T3 and CK3 were subjected to the same treatment of enzyme inactivation (220 °C, 8 min), rolling (12 min), and drying (primary drying at 125 °C/1.5 h, cooling for 2 h, final drying at 85 °C/2.5 h). We obtained the corresponding raw oolong tea from T3 (Rt) and CK3 (Rc) (Figure 2).

All the above treatments were carried out in a ventilated room with a humidity of 48%, a temperature of 25 °C, and a grade of 3 to 4 on the southeast wind scale [8]. The sampling for each treatment was repeated three times. All samples were wrapped in tin foil, fixed via the liquid nitrogen sample-fixing method, and placed in a −70 °C refrigerator.

### 2.2. Extraction and Detection of Volatile Compounds

In total, 2.0 g of tea power was weighed and immediately transferred to a 20 mL headspace vial; then, the volatile compounds were extracted via the HS-SPME (headspace solid-phase microextraction) method and detected via GC-TOF-MS (gas chromatography–time-of-flight–tandem mass spectrometry), using ethyl decylate (Sigma, St. Louis, MO, USA) as an internal standard. The SPME conditions were: the extraction needle PDMS/DVB9 was set with an incubation temperature of 80 °C, incubation time of 31 min, extraction time of 60 min, and resolution time of 3.5 min. The chromatograph conditions were: A 30 m × 0.25 mm × 0.25 μm chromatographic column (Rxi-5silMS) was used, the temperature of the sample inlet was controlled at 250 °C, the temperature of the transmission line was controlled at 275 °C, helium was used as the carrier gas, and the helium flow rate was set to 1 mL/min. The programmed heating process was maintained at 50 °C for 5 min, which rose to 210 °C at a rate of 3 °C/min for 3 min, and then rose again to 230 °C at a rate of 15 °C/min with non-shunt injection. The mass spectrometry conditions were: the solvent delay time was controlled within 300 s, the scanning range was controlled between 30 and 500 amu, the collection rate was 10 specs/s, the detector voltage was 1530 V, the EI ionization energy was 70 eV, and the ion source temperature was 250 °C. The data of the measured volatile metabolites were analyzed and processed using the instrument’s built-in software and then compared to the mass spectrometry data found in the National Institute of Standards and Technology (NIST) database, searching for relevant mass spectrometry data, and analyzing the base peaks, mass-to-nucleus ratios, and relative kurtosis. Finally, each peak based on the retention time and mass spectrometry was confirmed, VB/VP components were identified using peak areas of more than 1% as effective observation values, and the percentage content of aromatic compounds in each sample using the peak area normalization method was determined [22,24].

The OAV (odor activity value) is usually applied to measure the contribution degree of aroma compounds. Compounds with an OAV of more than 1 are considered key aroma compounds in oolong tea production. The OAV is calculated using the following formula:$$OAV=\frac{Cx}{OTx}$$
where *Cx* is the concentration of each volatile compound (μg·L^−1^), and *OTx* is the aroma threshold of the volatile compounds in water (μg·L^−1^) [35].

### 2.3. Genome and Transcriptome Data Sources and Screening

By combining the Tea Plant Information Archive (TPIA, http://tpia.teaplant.org/ accessed 28 April 2023) with a transcriptome database of oolong tea processing samples, we found 16 transcripts of *CsAAAT*-related genes in metabolic pathways of aromatic compound formation. The *CsAAAT1 *(MH544095) and *CsAAAT2* (MH544096) gene sequences were both sourced from the GenBank database on the NCBI website (https://www.ncbi.nlm.nih.gov/genbank/, accessed date 20 April 2023).

### 2.4. Construction of Phylogenetic Evolutionary Tree

For the 16 selected members of *CsAAAT* and the nucleic acid sequences of *CsAAAT1* and *CsAAAT2*, Mega 6.0 and the maximum likelihood (ML) tree method were applied to construct the phylogenetic evolutionary tree, and further phylogenetic analysis was carried out, which was saved as an NWK document. Finally, further visual editing was conducted on the following website: http://itol.embl.de/, accessed date 20 April 2023.

### 2.5. Extraction of Total RNA and Synthesis of cDNA

According to the previous experimental methods used by our research group, the total RNA during the processing of the tea leaves, including the fresh leaves, withered leaves, and experimental and control groups, was extracted using the centrifugal column method with the RNAprep Pure Plant Kit (Tiangen Biotech Co., Ltd., Beijing, China), following the manufacturer’s instructions. The concentration and purity of the RNA were detected using an ultra-micro nucleic acid analyzer, and the integrity was detected via 1.2% agarose gel electrophoresis. Finally, the cDNA was synthesized with 1000 ng of total RNA via reverse transcription using the PrimeScript RT Reagent Kit with a gDNA Eraser (TaKaRa Biotech Co., Ltd., Dalian, China) and stored at −20 °C for future use.

### 2.6. qRT-PCR Conditions

The SYBR @ Premix Ex TaqTM reagent kit was used to detect the expression of key *CsAAAT* genes during the processing of oolong tea. The specific primers for the target gene designed using DNAMAN are shown in Appendix A, and the specificity of the qRT-PCR amplicons was confirmed via sequencing. *CsGAPDH* (KA295375.1) was used as the internal reference gene. The total volume of the reaction system was 200 μL, including 10.0 μL of SYBR Green Master Mix, 0.4 μL each of forward and reverse primers (the concentration of all primers was 10 μmoL/μL), 2.0 μL of cDNA template, 0.4 μL of passive reference dyes, and 6.8 μL of sterile deionized water. The qRT-PCR conditions were 30 s at 95 °C, and the amplification program conditions were as follows: 94 °C for 30 s, 94 °C for 5 s, and 60 °C for 30 s, performing 40 cycles, followed by 60 °C to 95 °C melting curve detection (Appendix A), and a final step of 10 min at 72 °C for extension. After that, ABI QuantStudio 3 was used to measure the expression of various target genes in the samples, and the 2^-ΔΔCt^ method was applied to calculate the relative expression levels of relevant genes [36]. Fresh tea leaves (F) were used as the control tissue, where the relative expression was set to 1.0 (2^^0^).

### 2.7. Data Analysis

PASW statistics 18.0 was used for the analysis, and differences in the content of VBs/VPs and the relative expression levels of *CsAAAT* genes were analyzed using the Tukey’s honest significant difference (HSD) test [10] at significance levels of *p* < 0.05 and *p* < 0.01. Correlations between the content of VBs/VPs and the relative expression levels of *CsAAAT*-related genes in the experimental and control groups were analyzed using Spearman correlation analysis. GraphPad Prism 6.0 and TBtool were used for analysis and drawing.

## 3. Results

### 3.1. Analysis of the Dynamic Changes in VBs/VPs during the Post-Harvest Process of Oolong Tea

A total of 12 types of VBs, including benzaldehyde, benzyl alcohol, benzyl isobutyrate, and methyl formate, were detected in the samples. It was found that the content of benzaldehyde significantly increased after the turnover treatment (*p* < 0.01) and reached its maximum (0.02%) in T3. Although the content of benzyl alcohol increased during the post-harvest process of oolong tea, its content in T3 was lower than that in CK3 (Figure 3).

We measured four types of VPs that showed different change patterns, namely phenylethanol, 2-phenylethyl butyrate, methyl phenylacetate, and methyl phenylacetate. The contents of phenylethanol and 2-phenylethyl butyrate progressively increased as the post-harvest process moved forward, reaching the maximum value in T3, and showed a highly significant difference when compared to CK3 (*p* < 0.01). Methyl phenylacetate increased after the solar-withering treatment and decreased after the turnover treatment, while the content of phenylethyl acetate changed in the opposite direction (Figure 4).

### 3.2. Evaluation of Total RNA Quality

The centrifugal column method was used to purify the total RNA in the samples. The results of the total RNA quality detection showed that the concentrations of all samples were between 400 and 800 ng/mL, and the A260/A280 ratios ranged from 2.00 to 2.10 (Appendix A). The results of the agarose gel electrophoresis showed that the 18S bands and 28S bands of each sample were clear, uniform, and without trailing edges (Appendix A). Overall, a good purity and integrity were preserved during the solar-withering and turnover treatments, although the fresh tea leaves were in an in vitro state for a period of time. This indicated that the level of gene transcription was still present, which laid the foundation for the following reverse transcription quantitative real-time PCR (RT-qPCR) assay, as follows.

### 3.3. Screening and Analysis of CsAAAT Genes during the Post-Harvest Process of Oolong Tea

Based on the transcriptome database of oolong tea processing samples generated by the research group [10,37], a local BLAST operation was performed on the nucleic acid sequences of the *CsAAAT1* (Accession No. MH544095) and *CsAAAT2* genes (Accession No. MH544096). Nine highly similar sequences to the *CsAAAT1* gene were screened (>99%, E_value = 0), tentatively named *CsAAAT1-1* to *CsAAAT1-9*, whose sequence lengths ranged from 567 bp to 1422 bp. Similarly, eight highly similar sequences to *CsAAAT2* were screened (>99%, E_value = 0), tentatively named *CsAAAT2-1* to *CsAAAT2-8*, and the sequence lengths ranged from 1053 bp to 1572 bp (Table 1). We named these two groups the *CsAAAT1*-type genes and the *CsAAAT2*-type genes. The nucleic acid sequences of these two groups of genes are presented in Appendix A. The phylogenetic tree constructed based on the sequence homology of the protein sequences of *Cs*AAAT1 and *Cs*AAAT2 found that two types of genes could be clustered into one cluster (Figure 5).

The fragments per kilobase per million (FPKM) of 16 candidate *CsAAAT* genes was analyzed using a cluster heatmap. The results showed that the expression values of the *CsAAAT* genes could be divided into three groups: group A, group B, and group C. Group A contained seven genes. As for the FPKM in this group, the values of *CsAAAT1-1*, *CsAAAT1-3*, and *CsAAAT1-9* in T3 were significantly higher than those in CK3 (*p *< 0.01), while the value of *CsAAAT2-3* in T3 was significantly higher than that in CK3 (*p *< 0.05). There were three genes in group B, and the FPKM of this group generally decreased after the solar-withering treatment, but *CsAAAT2-5* and *CsAAAT2-1* were upregulated after the turnover treatment, and the difference in the FPKM between T3 and CK3 was highly significant (*p *< 0.01). The overall expression value trends of six genes in group C were similar to those in group A, but the expression abundance was lower (Figure 6).

We used qRT-PCR to verify the expression level of six *CsAAAT* genes. The results are shown in Figure 7, where the *CsAAAT* genes present a convergent expression pattern. In other words, as the manufacturing progressed, the *CsAAAT* gene expression level was significantly upregulated, reaching its maximum in T3, and there was a significant difference between T3 and CK3. It is worth noting that the solar-withering treatment significantly promoted the expression level of the *CsAAAT2*-type genes, while in the *CsAAAT*1-type genes, the change in the expression level of fresh leaves after solar-withering was not significant. Correlation analysis showed that there was a significant positive correlation between the relative expression of the *CsAAAT* genes and the FPKM values (*r* > 0.70) (Table 2), which confirmed that the key *CsAAAT* genes regulated the formation of VBs/VPs during the post-harvest process of oolong tea.

### 3.4. Correlation Analysis between Dynamic Changes in VB/VP Content and CsAAAT Gene Expression during the Post-Harvest Process of Oolong Tea

Through heatmap analysis of the correlation between the VB/VP content and the *CsAAAT* gene expression level during the post-harvest process of oolong tea, we found that the correlation between the VB/VP content and the *CsAAAT* gene expression level was mainly positive (142 pairs, 68.27%), with 80 pairs (56.34%) achieving a significant positive correlation, while the negative correlation reached a significant level, with only 4 pairs (6.06%) (Figure 8A). Statistically, there were significant (*p* < 0.05) or extremely significant (*p* < 0.01) positive correlations between benzyl alcohol and phenylethanol and the *CsAAAT* gene, with 12 and 14, respectively. Among the *CsAAAT1*-type genes, the number of significant and above correlations between the *CsAAAT1-3* and *CsAAAT1-5* genes and VBs/VPs was nine; among the *CsAAAT2*-type genes, there was a significant or extremely significant positive correlation between the *CsAAAT2-3* gene and 10 VBs/VPs (Figure 8B). In combination with the cluster analysis of the FPKM of the genes, it was found that there were three *CsAAAT1*-type genes and three *CsAAAT2*-type genes closely related to VB/VP formation, namely, *CsAAAT1-1*, *CsAAAT1-3*, and *CsAAAT1-9*, and *CsAAAT2-1*, *CsAAAT2-3*, and *CsAAAT2-6*, respectively. To sum up, the upregulation of the expression of key *CsAAAT* genes had significant synergy with the accumulation of VB/VP content.

### 3.5. Detection and Analysis of VBs/VPs in Raw Oolong Tea

Sensory evaluation was employed for Rt and Rc, which were produced from T3 and CK3 under a uniform manufacturing process. The results showed that the aroma of Rt was fragrant and lasting, with a floral character, basically conforming to the main quality characteristics of oolong tea; meanwhile, the aroma of Rc exhibited clean and refreshing sensory characteristics, but with a less gentle, flowery character (Appendix A). Targeted analysis of VB and VP components in raw oolong tea revealed that five types of VBs and three types of VPs were detected in Rt and Rc. Among them, benzoic acid, 2-hydroxy-, pentyl ester was only detected in Rt, and this component was not detected in T3 and R3 (Figure 9A). Through differential analysis, it was found that the content of VBs/VPs in Rt was significantly higher than that in Rc, except for methyl phenacylacetate (Figure 9B). Based on the calculation of the OAVs of the VBs/VPs in raw oolong tea, combined with Table 3, 2-hydroxy benzoic acid methyl ester and phenol alcohol both had OAVs greater than 1.0 in Rt and Rc. However, the OAV of 2-hydroxy benzoic acid methyl ester in Rt was more than two times that in Rc, while the OAV of benzyl alcohol was greater than 1.0 in Rt.

## 4. Discussion

### 4.1. Dynamic Changes in VBs/VPs in Oolong Tea Products

VBs/VPs play an important role in the formation of oolong tea’s natural floral and fruity aroma [12]. Benzaldehyde, a key intermediate of VBs, gradually decreased in content during the solar-withering stage, but after the turnover treatment, its content significantly increased (*p* < 0.01), indicating that the accumulation of benzaldehyde had no significant effect under light or heat stress during the solar-withering process but significantly responded to mechanical stress during turnover. However, the dynamic change in the benzyl alcohol content during the solar-withering stage was exactly the opposite of that in the benzaldehyde content, which conformed to the transformation rule of trans-cinnamic acid (PAL branch) and indirectly indicated that the solar-withering and turnover treatments were effective measures for gradually promoting the formation of VBs in oolong tea. However, Chen S et al. (2020) and Liu PP et al. (2018) pointed out that the content of benzaldehyde and benzyl alcohol did not significantly change during the manufacturing process of Tieguanyin oolong tea [24,38]. This result was different from our results, which might be due to the differences in tea varieties and the intensity of the turnover treatment. Oolong tea made from the ‘Huangdan’ variety is famous for its strong aroma, while the ‘Tieguanyin’ variety has a more peaceful aroma. Wang Z Q et al. (2015) found that the content of benzyl alcohol, phenylethanol, and phenylacetaldehyde in Huangdan black tea was higher than that in Tieguanyin black tea [39]. These studies suggest that the quality characteristics of the strong aroma of Huangdan oolong tea are related to the accumulation of VBs/VPs during manufacturing. More importantly, the content of benzyl alcohol in T3 was higher than that in CK3, indicating that the spreading treatment was an effective way to facilitate the accumulation of benzyl alcohol. On the contrary, in terms of VBs, there was generally a significant increase in T3, and a significant difference compared to CK3. Cho et al. (2007) found that the content of benzaldehyde in tea leaves treated with turnover treatments with 5 repetitions increased by 236 times, and the content of benzyl alcohol and phenethyl alcohol also increased by 38 times and 41 times, respectively [40], in line with our results. Therefore, it is speculated that further aggravation of the mechanical stress of turnover is an effective measure for promoting the generation and accumulation of VBs/VPs.

### 4.2. Expression Analysis and Regulation of Key CsAAAT Genes during the Post-Harvest Process of Oolong Tea

Upregulation of key structural genes in metabolic pathways is a prerequisite for the formation and accumulation of VBs/VPs [41]. We found that both the *CsAAAT1* and *CsAAAT2* genes had a significantly upregulated expression during the post-harvest process of oolong tea, and there was an obvious stress response to the exogenous abiotic stress of the mechanical force of turnover. Combined with the phylogenetic tree of the *CsAAAT* genes, the result showed that there is a highly conservative type in the sequence of the aromatic amino acid aminotransferase gene family in tea plants [42], along with a convergence effect on the expression patterns. It is important to highlight that methyl phenylacetate was the only VP that was not significantly correlated with all *CsAAAT* genes, which might indicate a relationship between the formation of this substance and the PAAS branch, rather than the AAAT branch. There was a generally strong positive correlation between VBs and key *CsAAAT* genes, suggesting a convergence effect in the formation of VBs both in the PAL and AAAT branches during the post-harvest process of oolong tea. This result agrees well with previous studies. For example, Wang et al. (2018) reported the expression patterns of *CsAAAT* genes under pre-harvest treatment of fresh tea leaves. There was no significant change in the expression level of *CsAAAT1*-type genes during solar-withering, but the expression levels of three *CsAAAT2*-type genes in SW were significantly higher than those in F, indicating that the *CsAAAT1* gene might be dominant in response to mechanical stress caused by turnover, while the *CsAAAT2* gene may respond to not only mechanical stress but also light and heat stress caused by solar-withering treatment. This is basically consistent with the viewpoint proposed by Yu XL (2021) that “yellow light treatment could promote the metabolism of VBs/VPs during the spreading treatment, a process similar to solar-withering, which enhanced the formation of high aroma quality of green tea” [43]. Furthermore, our results are similar to those reported by Hirata H et al. (2016), who found that “the petals of roses activate AAAT to catalyze L-phe synthesis of phenylpyruvate under high temperature stress in summer” [44]. The low sensitivity of *CsAAAT1*-type genes to temperature might be related to the spatial expression differences in this type of gene. Therefore, we speculated that the expression abundance of *CsAAAT1*-type genes might be lower than that of *CsAAAT2*-type genes under heat stress during the post-harvest process of oolong tea.

After the turnover treatment, VBs/VPs changed and volatilized under green-killing and rolling treatments, and finally, only four types of VBs and three types of VPs were left in the raw oolong tea. Interestingly, benzoic acid, 2-hydroxy-, pentyl ester was only identified in Rt, indicating that after drying, the VB components in the tea leaves were still changing and forming through non-enzymatic conversion, which would eventually contribute to the aromatic quality of oolong tea, to some extent. These results are in line with those of a previous study [45], wherein short-term solar-withering facilitated the accumulation and release of volatile components. As for benzyl alcohol, phenyl ethanol, and methyl salicylate, these components all contributed to a long-lasting floral fragrance in Rt, similar to previous studies.

## 5. Conclusions

In this research, the relative content of VBs/VPs was quantitatively analyzed using GC-TOF-MS technology, and the key VB/VP components contributing to the aromatic quality of oolong tea were identified based on the OAV calculation. Furthermore, the key *CsAAAT* genes were screened by combining a tea genome database and a transcriptome database of oolong tea processing samples. The following conclusions were drawn:
(1)A total of 13 types of VBs/VPs were measured in the samples, including 9 types of VPs and 4 types of VBs, and the mechanical force of turnover was the key exogenous stress inducing VBs/VPs (excluding methyl phenylacetate).(2)The OAVs of 2-hydroxy benzoic acid methyl ester and phenylethyl alcohol were greater than 1.0; therefore, these specific VB/VP components could be considered key components contributing to the aromatic quality of oolong tea.(3)Six key *CsAAAT* genes closely related to VB/VP formation were screened, namely, *CsAAAT1-1*, *CsAAAT1-3*, *CsAAAT1-9*, *CsAAAT2-1*, *CsAAAT2-3*, and *CsAAAT2-6.*(4)The upregulation of the expression level of *CsAAAT2*-type genes may respond to light stress during solar-withering, in addition to the induction of the mechanical force of turnover.

## Figures and Tables

**Figure 1 metabolites-13-00868-f001:**
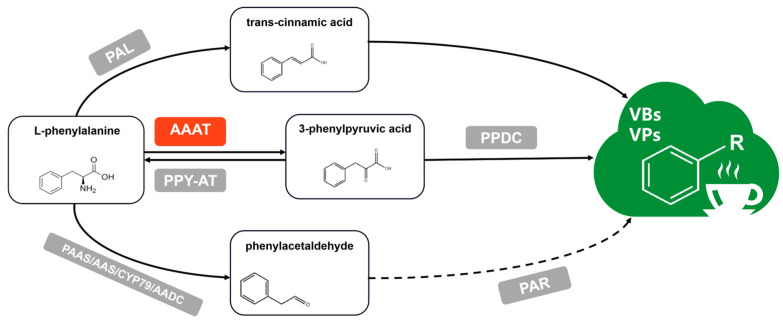
Three pathways of VB/VP formation in tea plants [18,26,27,28,29,30,31,32,33]. Potential metabolic pathways for converting L-phenylalanine into VBs/VPs in plants. Solid arrows indicate proven metabolic processes, while dashed arrows indicate unproven metabolic processes. The enzymes involved in the corresponding metabolic process are shown in the box, and the numbers in parentheses indicate the source of the literature. PAL: L-phenylalanine ammonia lyase; AAAT: aromatic amino acid aminotransferase; PPY-AT: phenylpyruvate aminotransferase; PDDC: phenylpyruvate decarboxylase; PAAS: phenylacetaldehyde synthase; AAS: aromatic aldehyde synthase; CYP79: cytochrome P450 family 79 enzyme; AADC: aromatic amino acid decarboxylase; PAR: phenylacetaldehyde reductase.

**Figure 2 metabolites-13-00868-f002:**
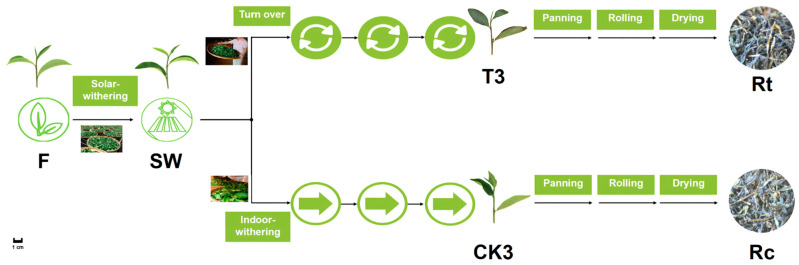
Manufacturing process diagram of fresh tea leaves. The manufacturing process steps of fresh tea leaves (F) and solar-withered leaves (SW). The upper branch of the diagram corresponds to the experimental group, which consisted of the turnover treatment repeated three times (T3), while the lower branch corresponds to the control group that was spread without turnover treatment (CK3), and Rt and Rc indicate the raw oolong teas made from samples T3 and CK3, respectively. Every sample was taken at the same time point as those in the experimental group. For the chemical profiling of VBs/VPs, the relevant samples were F, SW, T3, CK3, Rt, and Rc; for the *CsAAAT* gene expression profiling, the relevant samples were F, SW, T3, and CK3. Each sample was immediately frozen without panning.

**Figure 3 metabolites-13-00868-f003:**
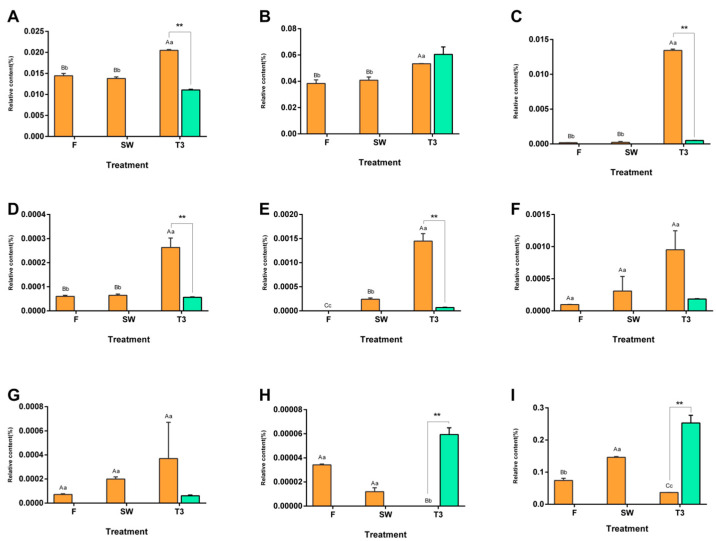
Dynamic changes in volatile benzenoids (VBs) during the manufacturing process of oolong tea. The dynamic change in VBs on the main process nodes during the processing of oolong tea. The numerical value represents the peak area abundance accumulated by the compounds. (**A**) Benzaldehyde, (**B**) benzyl alcohol, (**C**) benzyl isobutyrate, (**D**) methyl benzoate, (**E**) benzyl butyrate, (**F**) benzyl propionate, (**G**) benzyl formate, (**H**) 2-hydroxy-, 3-hexenylester, (z)-benzoicaci, and (**I**) 2-hydroxy benzoic acid methyl ester. Note: Different uppercase (A, B, C) and lowercase letters (a, b, c) represent significant differences at *p* < 0.01 and *p* < 0.05, respectively, while ** represent significant differences in the content at *p* < 0.05 and *p* < 0.01, respectively, between the turned-over leaves and the spread leaves.

**Figure 4 metabolites-13-00868-f004:**
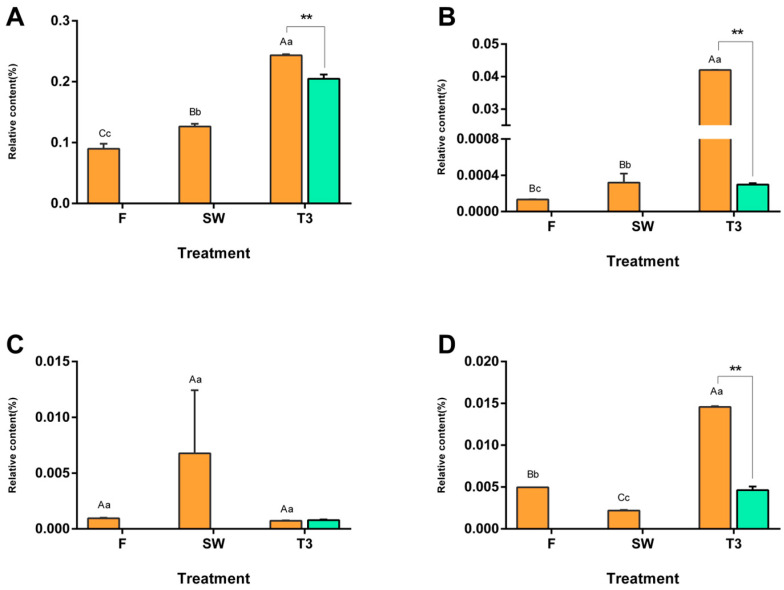
Dynamic changes in volatile phenylpropanoids (VPs) during the manufacturing process of oolong tea. The dynamic change in VPs on the main process nodes during the processing of oolong tea. The numerical value represents the peak area abundance accumulated by the compounds. (**A**) Phenethyl alcohol, (**B**) phenethyl butyrate, (**C**) methyl phenylacetate, and (**D**) phenethyl acetate. Note: Different uppercase (A, B, C) and lowercase letters (a, b, c) represent significant differences at *p* < 0.01 and *p* < 0.05, respectively, while ** represent significant differences in the content at *p* < 0.05 and *p* < 0.01, respectively, between the turned-over leaves and the spread leaves.

**Figure 5 metabolites-13-00868-f005:**
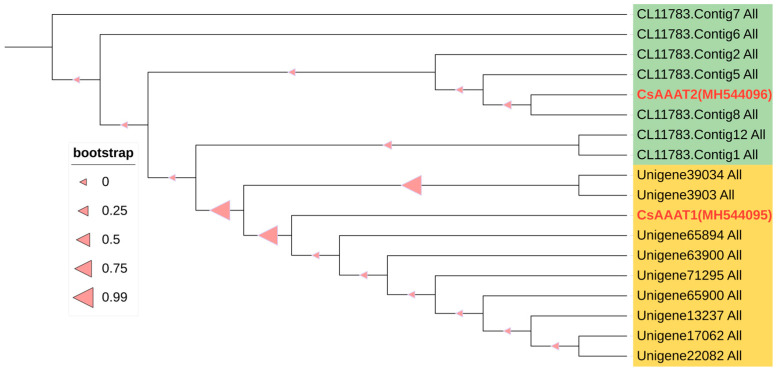
Phylogenetic tree based on protein sequence homology of the *CsAAAT1* and *CsAAAT2* genes. Based on the transcriptome database of the oolong tea processing samples, a phylogenetic tree of the *CsAAAT1* and *CsAAAT2* genes, registered in GenBank, was constructed with a bootstrap value of 1000. The light-yellow- and light-green-colored blocks are used to distinguish between the homologous genes of the *CsAAAT1* and *CsAAAT2* genes, respectively, and the registered genes are labeled with red dots.

**Figure 6 metabolites-13-00868-f006:**
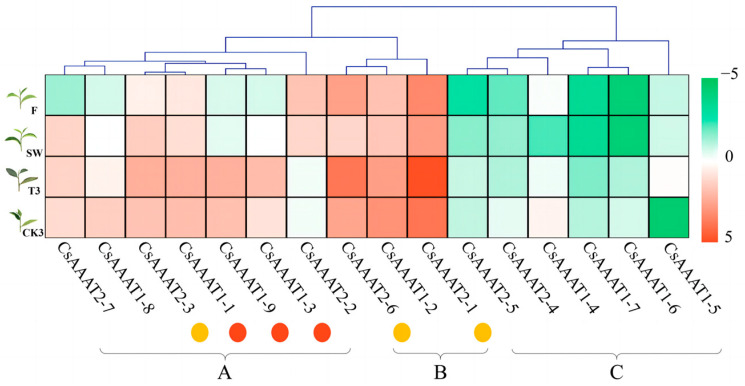
FPKM of *CsAAAT* gene transcripts during the manufacturing process of oolong tea based on transcriptome data. Based on the transcriptome data of the oolong tea processing samples, the figure shows the heatmap and the maximum likelihood tree cluster analysis of the FPKM values of the fresh leaves (F), solar-withered leaves (SW), turnover treatment repeated three times (T3), and the control group without turnover treatment repeated three times (CK3). According to the clustering results, three groups were preliminarily established: A, B, and C. Light-red and light-yellow dots are used to mark the different expression genes of *CsAAAT1* and *CsAAAT2* in groups A and B, respectively.

**Figure 7 metabolites-13-00868-f007:**
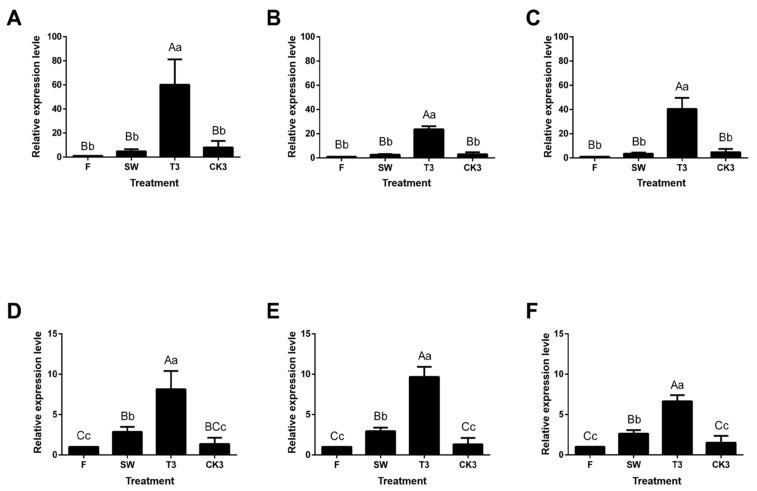
RT-qPCR verification of key *CsAAAT* genes during the post-harvest process of oolong tea. The relative expression levels of key *CsAAAT* genes obtained through screening in the fresh tea leaves (F), solar-withered leaves (SW), turnover treatment repeated three times (T3), control group without turnover treatment (CK3), and fresh leaves were used as control tissue, where the relative expression was set to 1.0 (2^^0^). (**A**) *CsAAAT1-1* gene, (**B**) *CsAAAT1-3* gene, (**C**) *CsAAAT1-6* gene, (**D**) *CsAAAT1-1* gene, (**E**) *CsAAAT1-3* gene, and (**F**) *CsAAAT1-6* gene. Note: Different uppercase (A, B, C) and lowercase letters (a, b, c) represent significant differences at *p* < 0.01 and *p* < 0.05, respectively.

**Figure 8 metabolites-13-00868-f008:**
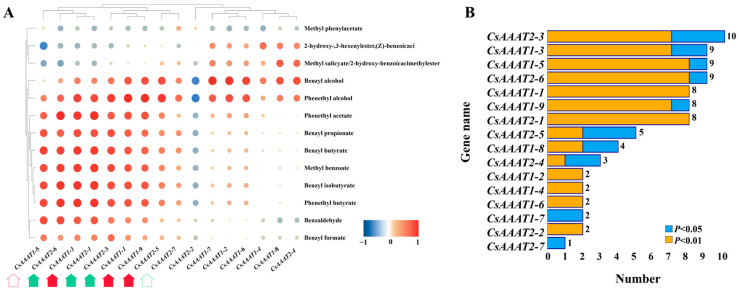
Correlation analysis between FPKM of *CsAAAT* genes and VB/VP content. (**A**) Heatmap of the correlation coefficient between the FPKM of *CsAAAT* genes and VB/VP content during the manufacturing process of oolong tea. Positive correlations are represented by red dots. The stronger the correlation, the larger the area and the darker the color of the red dots. Conversely, blue represents weaker correlations. The arrows below the genes indicate candidate genes with a significant correlation with VBs/VPs, where those consistent with the results of the cluster screening are indicated by a solid arrow; otherwise, a hollow arrow is used. (**B**) Based on the results of the correlation coefficient heatmap, the figure shows a statistical analysis of VBs/VPs with a correlation with *CsAAAT* genes during the manufacturing process of oolong tea. The yellow and blue columns represent the number of extremely significant (*p* < 0.01) and significant (*p* < 0.05) correlations between the *CsAAAT* genes and the VBs/VPs, respectively.

**Figure 9 metabolites-13-00868-f009:**
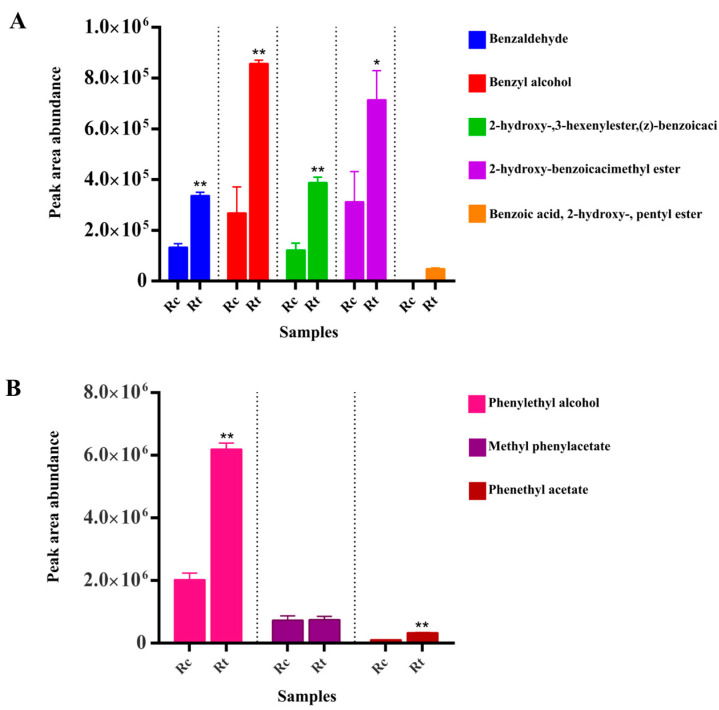
Comparative analysis of volatile benzenoids and volatile phenylpropanoids in raw oolong tea. (**A**) Peak area abundance of volatile benzenoids (VBs) in raw oolong tea under different treatments. (**B**) Peak area abundance of volatile phenylpropanoids (VPs) in raw oolong tea under different treatments. Black dashed lines are used to distinguish between different VBs/VPs in raw tea. Rt denotes raw oolong tea from T3 in the experimental group, and Rc denotes raw oolong tea from CK3 in the control group. *, ** Represent significant differences between the relative expression levels and FPKM values of key transcripts at *p* < 0.05 and *p* < 0.01, respectively.

**Table 1 metabolites-13-00868-t001:** Local BLAST results of *CsAAAT* genes based on a transcriptome database.

Gene Name	Transcript	Number	Sequence Length/bp	Score/Site	E Values	Identity
*CsAAAT1* (1266 bp)	Unigene22082_All	* CsAAAT1-1 *	1422	2486	0.0	1263/1266 (99%)
	Unigene17062_All	* CsAAAT1-2 *	1266	2478	0.0	1262/1266 (99%)
	Unigene13237_All	* CsAAAT1-3 *	1329	2242	0.0	1137/1139 (99%)
	Unigene65900_All	* CsAAAT1-4 *	1059	1744	0.0	886/888 (99%)
	Unigene71295_All	* CsAAAT1-5 *	903	1729	0.0	884/888 (99%)
	Unigene65894_All	* CsAAAT1-6 *	1059	1697	0.0	880/888 (99%)
	Unigene3903_All	* CsAAAT1-7 *	858	1586	0.0	818/824 (99%)
	Unigene39034_All	* CsAAAT1-8 *	993	1586	0.0	818/824 (99%)
	Unigene63900_All	* CsAAAT1-9 *	567	1108	0.0	565/567 (99%)
*CsAAAT2* (1266 bp)	CL11783.Contig7_All	* CsAAAT2-1 *	1266	2478	0.0	1262/1266 (99%)
	CL11783.Contig1_All	* CsAAAT2-2 *	1572	2478	0.0	1262/1266 (99%)
	CL11783.Contig12_All	* CsAAAT2-3 *	1227	2401	0.0	1223/1227 (99%)
	CL11783.Contig2_All	* CsAAAT2-4 *	1167	2244	0.0	1144/1148 (99%)
	CL11783.Contig5_All	* CsAAAT2-5 *	1161	2060	0.0	1051/1055 (99%)
	CL11783.Contig8_All	* CsAAAT2-6 *	1026	1748	0.0	894/898 (99%)
	CL11783.Contig6_All	* CsAAAT2-7 *	1053	1628	0.0	833/837 (99%)

Nine highly similar sequences to the *CsAAAT1* gene and eight highly similar sequences to *CsAAAT2* were screened. The local alignment association information between each related sequence and the *CsAAAT1* and *CsAAAT2* genes is shown, mainly including score/site, E values, and identity.

**Table 2 metabolites-13-00868-t002:** Analysis of the correlation coefficient between the relative expression levels and FPKM values of key gene transcripts of *CsAAAT* during the post-harvest process of oolong tea.

*CsAAAT* Type	Gene Name	Transcript	Correlation Coefficient	Significance
*CsAAAT1*	*CsAAAT1-1*	Unigene22082_All	0.824	**
	*CsAAAT1-3*	Unigene13237_All	0.948	**
	*CsAAAT1-9*	Unigene63900_All	0.786	**
*CsAAAT2*	*CsAAAT2-1*	CL11783.Contig7_All	0.830	**
	*CsAAAT2-3*	CL11783.Contig12_All	0.851	**
	*CsAAAT2-6*	CL11783.Contig8_All	0.796	**

Evaluation of the correlation between the relative expression level of screened genes and transcriptome data. The larger the absolute value of the correlation coefficient, the higher the degree of correlation between the two. ** Indicates a highly significant positive correlation between the two.

**Table 3 metabolites-13-00868-t003:** The aroma character, concentration, OAV, and threshold of related VBs/VPs in raw oolong tea.

Types	Component	CAS	Character	Concentration (μg/g)		OAV		Threshold (ug/kg)
				Rt	Rc	Rt	Rc	-
VBs	Benzaldehyde	100-52-7	Bitter almond odor	10.74 ± 0.44	4.25 ± 0.45	0.68	0.27	350
	Benzyl alcohol	100-51-6	Rose flavor	27.38 ± 0.40	8.58 ± 3.28	1.51	0.47	400
	2-Hydroxy-, 3-hexenylester, (z)-benzoicaci	65405-77-8	Fresh fragrance	12.41 ± 0.65	3.92 ± 0.85	-	-	-
	2-hydroxy benzoic acid methyl ester	119-36-8	Fragrance of holly oil	22.80 ± 3.67	9.98 ± 3.80	12.56	5.50	40
	Benzoic acid, 2-hydroxy-, pentyl ester	2050-08-0	Orchid fragrance	1.55 ± 0.10	0.00	-	-	-
VPs	Phenylethyl alcohol	60-12-8	Rose-like	197.89 ± 6.10	64.71 ± 6.80	5.81	1.90	750
	Methyl phenylacetate	101-41-7	Sweet floral fragrance	23.59 ± 4.17	24.05 ± 3.37	0.52	0.53	1000
	Phenethyl acetate	103-45-7	Rose-like	10.69 ± 0.15	3.45 ± 0.07	-	-	-

Comparative analysis of the concentrations and aroma contributions of 5 VBs and 3 VPs detected in raw tea. VBs: volatile benzenoids; VPs: volatile phenylpropanoids; CAS: Chemical Abstracts Service; Rt: raw oolong tea from T3 in the experimental group; Rc: raw oolong tea from CK3 in the control group; OVA: odor activity value. Threshold values are mainly taken from http://www.odour.org.uk/lriindex.html, accessed 20 May 2023. “-” Indicates that the threshold of the substance is still unknown.

## Data Availability

Data are contained within the article or the Appendix A.

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
