# Peer review of "The Dynamic Change in Aromatic Compounds and Their Relationship with CsAAAT Genes during the Post-Harvest Process of Oolong Tea"

_metabolites, 2023, doi:10.3390/metabo13070868_

Round 1

Reviewer 1 Report

The manuscript by Zi-Wei et al describes the influence of different post-harvest processes on the aromatic volatiles of oolong tea. The novel aspect of the research is the formation of volatile benzenoids and phenylpropanoids via the aromatic amino acid aminotransferases. The combination of metabolomics and transcriptomics support the presented conclusion about the aroma formation based on the manufacturing process applied.

The majority of legends needs to be rewritten in a scientific format, especially Table legends. The legend should describe everything you need to know for interpretation of the table/figure, without having to read the results section. The legend should be written in a sentence structure and not a list of e.g. genes/metabolites.

What difference does significant and extremely significant make to the results? How relevant is this in a biological context?

The manuscript is outlined very well, however it needs to be corrected by a native English speaker, especially for the descriptors of tea aroma. I am not sure what the authors mean with high and peaceful. Also, the authors should check the spelling of words (singular/plural) and capital/small letters. Quite a few of the sentences in the discussion are missing verbs. Be consistent with italics of gene names. Look into placing of spaces e.g. Line 230 and 233.

Author Response

Q1: The majority of legends needs to be rewritten in a scientific format, especially Table legends. The legend should describe everything you need to know for interpretation of the table/figure, without having to read the results section. The legend should be written in a sentence structure and not a list of e.g. genes/metabolites.

Response: Thank you for your reminder. We have rewritten the legends of most of the tables and figures, including Fig 1, Fig 2, Fig 3, Fig 4, Fig 7, Fig 9, Table 1, Table 2, and Table 3. We mainly referred to the scientific format of the latest papers in the journal of “Matabolites”[1-3].

Ref:

[1] Katsaounou, K.; Yiannakou,D.; Nikolaou, E.; Brown, C.;Vogazianos, P.; Aristodimou, A.; Chi, J.; Costeas, P.; Agapiou, A.; Frangou,E.; et al. Fecal Microbiota and Associated Volatile Organic Compounds Distinguishing No-Adenoma from High-Risk Colon Adenoma Adults. Metabolites 2023,13, 819.

[2] Zhang, Z.X.; Mo, R.M.;Liu, D.B.; Liu, Y.S.; Liu, C.H.; Li,Y.S.; Liu, Z.H.; Qin, D. Research on the Efficacy of Ganpu Vine Tea in Inhibiting Uric Acid Production. Metabolites 2023, 13, 704.

[3] Chang, L.; Zhou, G.; Xia, J. mGWAS-Explorer 2.0: Causal Analysis and Interpretation of Metabolite–Phenotype Associations. Metabolites 2023, 13, 826.

Q2: The manuscript is outlined very well, however it needs to be corrected by a native English speaker, especially for the descriptors of tea aroma.

Response: Thank you for your reminder. We have submitted the manuscript to English editing department of MDPI for language polishing. We hope that the polished manuscript could meet your requirements.

Q3: I am not sure what the authors mean with high and peaceful.

Response: Thank you for question. “high” and “peaceful”are vocabulary for tea sensory evaluation[1-2], used to indicate that oolong tea has high aroma quality. In detail, “high” is a manifestation of excellent and strong aroma quality; “peaceful” reflects the pure and normal aroma of tea.

Ref:

[1] GB/T 23776-2018, Methodology for sensory evaluation of tea[S]. Beijing: China Standard Press, 2018.

[2] GB/T 14487-2017, Tea vocabulary for sensory evolution[S]. Beijing: China Standard Press, 2017.

Q4: Also, the authors should check the spelling of words (singular/plural) and capital/small letters.

Response: Thank you for suggestion and reminder. We have checked the spelling of words (singular/literal) and capital/small letters for the whole manuscript.

Q5: Quite a few of the sentences in the discussion are missing verbs.

Response: Thank you for question. As for the discussion section, we checked each sentence and promptly supplemented and corrected the sentences which lacked the verbs.

Q6: Be consistent with italics of gene names. Look into placing of spaces e.g. Line 230 and 233.

Response: Thank you for suggestion and reminder. We have also checked and corrected the gene names in the whole manuscript, and gene names have been italicized. Names involving proteins have also been standardized and non italicized. In final, we have made adjustments to the placing of spaces in Line 230 and 233.

Reviewer 2 Report

The article is based mainly on the gene expression analysis and its correlation to the volatile benzenoids and phenylpropanoids. I do not see that the experimental set-up is proper enough to lead to a rigid conclusion.

The article's language is also poor and must be improved in term of tense, plural and singular verbs and many spelling mistakes.

In addition to mistakes in chemical names for example: 2-hydroxy-benzoicacimethyl (lines 19 and 437) and Line 56: phenylacetaldehyde synthetase (PAAS), it s synthase not synthetase...

oolong tea is Camellia sinensis. the scientific name should be indicated in the abstract so the gene initials  'Cs' are clear to the reader.

Line 154: the accession number for CsAAAT2 is missing as 'MH544095' is a repetition for CsAAAT1.

Line 157: CsAAAT is completely italic if referred to the sourced gene only (should be corrected in line 231). In section 2.3, CsAAAT should refer to the protein. So, only initials of the Latin name are italic. This error is repeated in the whole article. It cannot be that the NJ tree was really made of nucleic acid sequences as line 157 tells.

Section 2.3:  NJ tree is not as reliable as Maximum likelihood tree  in this case. NJ is based on genetic distance and is not likely to be affected by minimum change among CsAAAT sequences. Maximum likelihood should be constructed instead.

Line 169: How many ng of RNA was reverse transcribed to used whole 2 µl (line 177) of cDNA?

Line 175:NCBI accession number of CsGAPDH is missing.

Line 177: primers concentration is missing.

Line 178: which type of sterile water; distilled, deionized, etc? How was it produced?

Line 179: the programme is for normal PCR not qPCR. Also melt-curve is missing in the program.

Line 181: 2-ddCT requires a parallel expression study in control tissue, which is absent.

Lines 226 and 229: Neither the designed primers were tested for their specificity (no melt-curve) nor the sequences of the obtained amplicons were confirmed. This weakens the obtained results as the similarities among CsAAAT groups are >99%.

Section 3.3 should be shifted before section 3.2 and section 3.4 should be merged with 3.2.

Figure  1: the authors should  indicate the meant differences among dotted and solid arrows.

Lines 148, 149: The unit should be to the power (superscript) of -1

Line 60: first letter of the author name is capital not the whole name.

All supplementary data are missing and the visibility of the current figures is very weak.

The article's language is poor and must be improved in term of tense, plural and singular verbs and many spelling mistakes.

Author Response

Q1: The article is based mainly on the gene expression analysis and its correlation to the volatile benzenoids and phenylpropanoids. I do not see that the experimental set-up is proper enough to lead to a rigid conclusion.

Response: Thank you for your question. Our study was based on the confirmed AAAT pathway and the synthesis mechanism of pre-harvest related VBs/VPs in tea plants[1]. However, the formation and transformation mechanism of VBs/VPs, an important part of the aroma of oolong tea[2], is still unknown during the post-harvest process of oolong tea. Therefore, in our study, raw oolong tea and manufacturing samples were used as test materials, and targeted metabonomics combined with transcriptome analysis was also conducted. Through the result, we could better understand the formation mechanism of VBs/VPs during the post-harvest process of oolong tea based on the CsAAAT pathway by detecting the abundance of VBs/VPs and the relative expression level of selected CsAAAT genes during the post-harvest processing samples and raw oolong tea. Based on the similar research model, our research group has made some progress in the research of tea fatty acid metabolism pathway and tea aroma[3-5].

Ref:

[1] Wang,X.Q.; Zeng, L.T.; Liao, Y.Y.; Zhou, Y.; Xu, X.L.; Dong, F.; Yang, Z.Y. An alternative pathway for the formation of aromatic aroma compounds derived from L-phenylalanine via phenylpyruvic acid in tea (Camellia sinensis (L.) O. Kuntze) leaves. Food Chem. 2019, 270, 17-24.

[2] Yang, Z.Y.; Baldermann, S.; Watanabe, N. Recent Studies of the Volatile Compounds in Tea. Food Res. Int. 2013, 53, 585-599.

[3] Zhou, Z.W.; Wu, Q.Y.; Yao, Z.L.; Deng, H.L.; Liu, B.B.; Yue, C.; Deng, T.T.; Lai, Z.X.; Sun, Y. Dynamics of ADH and related genes responsible for the transformation of C6-aldehydes to C6-alcohols during the postharvest process of oolong tea. Food Sci. Nutr. 2020, 8, 104-113.

[4] Zhou, Z.W.; Wu, Q.Y.; Ni, Z.X.; Hu, Q.C.; Yang, Y.; Zheng, Y.C.; Bi, W.J.; Deng, H.L.; Liu, Z.Z.; Ye, N.X.; Lai, Z.X.; Sun, Y. Metabolic Flow of C6 Volatile Compounds from LOX-HPL Pathway Based on Airflow during the Post-harvest Process of Oolong Tea. Front. Plant Sci, 2021, 12, 738445-738445.

[5] Zhou, Z.W.; Wu, Q.Y.; Yang, Y.; Hu, Q.C.; Wu, Z.J.; Huang, H.Q.; Lin, H.Z.; Lai, Z.X.; Sun, Y. The Dynamic Change in Fatty Acids during the Postharvest Process of Oolong Tea Production. Molecules, 2022, 27, 4298.

Q2: The article's language is also poor and must be improved in term of tense, plural and singular verbs and many spelling mistakes.

Response: Thank you for your suggestion. The article has been submitted to English editing department of MDPI for language polishing. We hope that the polished text could meet your requirements.

Q3: In addition to mistakes in chemical names for example: 2-hydroxy-benzoicacimethyl (lines 19 and 437) and Line 56: phenylacetaldehyde synthetase (PAAS), it s synthase not synthetase...

Response: Thank you for your reminder. We apologize for our carelessness in writing, and we have corrected it in the manuscript.

Q4: oolong tea is Camellia sinensis. the scientific name should be indicated in the abstract so the gene initials  'Cs' are clear to the reader.

Response: Thank you for your suggestion. We have supplemented the scientific(Latin) name(Camellia sinensis) in the abstract (Line 13). The details were as follows:

“volatile benzenoids(VBs) and volatile phenylpropanoids(VPs) were essential aromatic components in oolong tea (Camellia sinensis). “

Q5: Line 154: the accession number for CsAAAT2 is missing as 'MH544095' is a repetition for CsAAAT1.

Response: Thank you for your reminder. We apologize for our carelessness. We have made corrections in the manuscript(Line 163).

Q6: Line 157: CsAAAT is completely italic if referred to the sourced gene only (should be corrected in line 231). In section 2.3, CsAAAT should refer to the protein. So, only initials of the Latin name are italic. This error is repeated in the whole article. It cannot be that the NJ tree was really made of nucleic acid sequences as line 157 tells.

Response: Thank you for your reminder. We have made corrections according to your request. For writing involving proteins, we have changed Cs to italics, while AAAT was not in italics. We hope our modification of gene and protein names could satisfy you. If necessary, we could make further modifications.

Q6: Section 2.3:  NJ tree is not as reliable as Maximum likelihood tree  in this case. NJ is based on genetic distance and is not likely to be affected by minimum change among CsAAAT sequences. Maximum likelihood should be constructed instead.

Response: Thank you for your suggestion. We have replaced the NJ tree with the Maximum Likelihood (ML) tree based on protein sequences through Mega V_7.0 software.(Figure 5)

Q7: Line 169: How many ng of RNA was reverse transcribed to used whole 2 µl (line 177) of cDNA?

Response: Thank you for your reminder. 1000 ng of total RNA was reverse transcribed to used whole 2 µL of cDNA. We have added it in the manuscript (section 2.5) 

Q8: Line 175:NCBI accession number of CsGAPDH is missing.

Response: Thank you for your reminder. CsGAPDH accession number is KA295375.1. We have added it in the manuscript(section 2.6).

Q9: Line 177: primers concentration is missing.

Response: Thank you for your reminder. primers concentration was 10 μmoL/μL. We have already added the concentration of primers in 2.6 section.

Q10: Line 178: which type of sterile water; distilled, deionized, etc? How was it produced?

Response: Thank you for your question. We obtained deionized water through high-pressure sterilization using a pressure cooker. (Line 189)

Q11: Line 179: the programme is for normal PCR not qPCR. Also melt-curve is missing in the program.

Response: Thank you for your reminder.  The program we used for qPCR does indeed have similarities with normal PCR, but we added 10.0 μL of SYBR Green Master Mix in the experimental system, running on  ABI QuantStudio 3(Thermo Fisher, USA) in the experimental system. In our previous studies, there were similar expressions[1,2]. If necessary, we could also make appropriate modifications to the title of “qPCR”. The melt-curve obtained from our experiment will be supplemented in the supplementary material.

[1] Zhou, Z.W.; Wu, Q.Y.; Ni, Z.X.; Hu, Q.C.; Yang, Y.; Zheng, Y.C.; Bi, W.J.; Deng, H.L.; Liu, Z.Z.; Ye, N.X.; Lai, Z.X.; Sun, Y. Metabolic Flow of C6 Volatile Compounds from LOX-HPL Pathway Based on Airflow during the Post-harvest Process of Oolong Tea. Front. Plant Sci, 2021, 12, 738445-738445. [2] Zhou, Z.W.; You, F.N.; Liu, B.B.; Deng, T.T.; Lai, Z.X.; Sun, Y. Effect of mechanical force during turn-over on the formation of aliphatic aroma in oolong Tea. Food Sci. 2019, 40, 52-59.

Q12: Line 181: 2-ddCT requires a parallel expression study in control tissue, which is absent.

Response: Thank you for your question. Indeed, the parallel expression study in control tissue was carried in fresh tea leaves(F). in other words, the relative expression level of F was 1.00.

Q13: Lines 226 and 229: Neither the designed primers were tested for their specificity (no melt-curve) nor the sequences of the obtained amplicons were confirmed. This weakens the obtained results as the similarities among CsAAAT groups are >99%.

Response: Thank you for your question. It was really a good question. Indeed, The specificity of the amplicon corresponding to the primers we designed was a challenge in this study. Although the identity were all 99%, this did not mean that the complete sequence (1 266 bp) between these candidate genes and the source gene(CsAAAT1 and CsAAAT2) has a 99% overlap. Identity only referred to the similarity of some segments, because the size of these candidate genes from the transcriptome was between 800-1 200 bp. What’s more, we also re-aligned the sequences of the amplicons, proving that all six pairs of primers had specific fragments.

Q14: Section 3.3 should be shifted before section 3.2 and section 3.4 should be merged with 3.2.

Response: Thank you for your suggestion. We have adjusted the order of the results section. We advanced section 3.3 to section 3.1, and merged sections 3.2 and section 3.4.

Q15: Figure  1: the authors should  indicate the meant differences among dotted and solid arrows.

Response: Thank you for your reminder. We re-write the legend of Figure 1, and the dotted and solid arrows were also explained in the legend (Line 74-77)

Q16: Lines 148, 149: The unit should be to the power (superscript) of -1

Response: Thank you for your reminder. We have superscript the -1 in the units.

Q17: Line 60: first letter of the author name is capital not the whole name.

Response: Thank you for your suggestion. We have changed the first letter of author's name to uppercase only.

Q18: All supplementary data are missing and the visibility of the current figures is very weak.

Response: Thank you for your suggestion. All supplementary data have been submitted previously. There may be a discrepancy between the manuscript for peer review and the submission system, but we could ensure that the supplementary data was certain. We have further improved the visibility of the current figures, and have already packaged and uploaded these image files 

Comments on the Quality of English Language

Q19: The article's language is poor and must be improved in term of tense, plural and singular verbs and many spelling mistakes.

Response: Thank you for your suggestion. We have submitted the manuscript to English editing department MDPI for language polishing. We hope that the polished text could meet your requirements.

Reviewer 3 Report

Generally, the presented manuscript metabolites-2474574 is interesting experimental work.

Unfortunately, I was not able to review the paper normally due to the low quality of the manuscript. Figures must be increased so that the text will be readable.

There are also technical mistakes (for instance, lines 45, 50, 149, etc.).

 Moderate editing of English language required.

Author Response

Generally, the presented manuscript metabolites-2474574 is interesting experimental work.

Unfortunately, I was not able to review the paper normally due to the low quality of the manuscript. Figures must be increased so that the text will be readable.There are also technical mistakes (for instance, lines 45, 50, 149, etc.).

Response: Thank you for your suggestion. We have submitted the manuscript to English editing department MDPI for language polishing.

We have further improved the visibility and quality of the current figures, and have already packaged and uploaded these image files. We hope that the polished manuscript and the improved figures could meet your requirements.

We have also actively verified and modified the technical mistakes on L45, L50, and L149.

Our study was based on the confirmed AAAT pathway and the synthesis mechanism of pre-harvest related VBs/VPs in tea plants[1]. However, the formation and transformation mechanism of VBs/VPs, an important part of the aroma of oolong tea[2], is still unknown during the post-harvest process of oolong tea. Therefore, in our study, raw oolong tea and manufacturing samples were used as test materials, and targeted metabonomics combined with transcriptome analysis was also conducted. Through the result, we could better understand the formation mechanism of VBs/VPs during the post-harvest process of oolong tea based on the CsAAAT pathway by detecting the abundance of VBs/VPs and the relative expression level of selected CsAAAT genes during the post-harvest processing samples and raw oolong tea. Based on the similar research model, our research group has made some progress in the research of tea fatty acid metabolism pathway and tea aroma[3-5].

Ref:

[1] Wang,X.Q.; Zeng, L.T.; Liao, Y.Y.; Zhou, Y.; Xu, X.L.; Dong, F.; Yang, Z.Y. An alternative pathway for the formation of aromatic aroma compounds derived from L-phenylalanine via phenylpyruvic acid in tea (Camellia sinensis (L.) O. Kuntze) leaves. Food Chem. 2019, 270, 17-24.

[2] Yang, Z.Y.; Baldermann, S.; Watanabe, N. Recent Studies of the Volatile Compounds in Tea. Food Res. Int. 2013, 53, 585-599.

[3] Zhou, Z.W.; Wu, Q.Y.; Yao, Z.L.; Deng, H.L.; Liu, B.B.; Yue, C.; Deng, T.T.; Lai, Z.X.; Sun, Y. Dynamics of ADH and related genes responsible for the transformation of C6-aldehydes to C6-alcohols during the postharvest process of oolong tea. Food Sci. Nutr. 2020, 8, 104-113.

[4] Zhou, Z.W.; Wu, Q.Y.; Ni, Z.X.; Hu, Q.C.; Yang, Y.; Zheng, Y.C.; Bi, W.J.; Deng, H.L.; Liu, Z.Z.; Ye, N.X.; Lai, Z.X.; Sun, Y. Metabolic Flow of C6 Volatile Compounds from LOX-HPL Pathway Based on Airflow during the Post-harvest Process of Oolong Tea. Front. Plant Sci, 2021, 12, 738445-738445.

[5] Zhou, Z.W.; Wu, Q.Y.; Yang, Y.; Hu, Q.C.; Wu, Z.J.; Huang, H.Q.; Lin, H.Z.; Lai, Z.X.; Sun, Y. The Dynamic Change in Fatty Acids during the Postharvest Process of Oolong Tea Production. Molecules, 2022, 27, 4298.

Round 2

Reviewer 2 Report

I thank the authors for their effort to improve the article. Now I can also see the submitted supplementary data. I strongly recommend the authors to carefully consider these points as they still affect the article methodology, hence the main conclusion:

qPCR program must have a phase at its end to record the melt-curve, which is still missing. Please, add it to the program and provide the melt-curve as mentioned in the authors' response.

Line 186: please add a statement that fresh tea leaves were used as control tissue where the relative expression was set to 2^0 (i.e. 1), also, in legend of figure 7.

The answer of Q13 is still unclear to me. If re-alignment of the amplicons was performed, the experimentally obtained qPCR amplicons should have been sequenced. If this is the case, please add a clear statement that specificity of the amplicons was confirmed via sequencing.

Please, provide the 9 CsAAAT1 and 8 CsAAAT2 sequences as supplementary data or provide their accession numbers if you have submitted them to NCBI.

Please, confirm that you have replaced figure 5 (and the upper display of Figure 6) with the newly constructed ML tree. It is uncommon to have 100% identical NJ and ML trees.

Line 179: The added accession number (KA295375.1) is for TSA. Fine, as the sequence still show 95% of a predicted CsGAPDH. I have copied NCBI sequence below because I cannot find the forward primer or the reverse complementary of the reverse primer shown in Table S1. This may affect the reliability of the chosen primers. 

>KA295375.1 TSA: Camellia sinensis tea_rep_c16164 mRNA sequence

GGCTCTGACAAAGAGACGATGTTGAGCTCGTCGCTGTCAACGACTCCCTTCATCACCACTGATTACATGACTTACATGTTCAAGTATGAAAAAAAAATGTGTTCACGGTCAATGGAAGCATCATGAACTCAAAGTGAAGGATTCCAAGACCCTTCTCTTTGGTGAGAAAGCAGTAGCTGTTTTTGGCCTTAGGAACCCAGAGGAGATCCCATGGGCTGAGACTGGAGCCGAATTCATTGTGGAGTCCACTGGTGTCTTCACTGATAAGGATAAGGCTGCTGCCCACTTGAAGGGTGGTGCAAAGAAGGTTATCATTTCAGCCCCTAGTAAGGATGCTCCCATGTTTGTTATGGGTGTCAATGAGAAGGATTACAAGCCAGATCTTCACATTGTTTCCAATGCTAGCTGCACTACCAACTGTCTTGCCCCCCTTGCTAAGGTTATCAATGACAAGTTTGTCATCGTTGAGGGTCTCATGACTACCGTGCACTCCATCACAGCCACACAAAAAACTGTTGATGGACCATCAAGCAAGGACTGGAGAGGTGGAAGAGCTGCTTCATTCAATATCATTCCTAGCAGCACTGGAGCTGCCAAGGCTGTTGGTAAGGTGCTGCCTGCACTTAATGGGAAATTGACTGGAATGGCTTTCCGCGTTCCCACTGTCGATGTCTCAGTGGTTGACCTCACTGTGAGGCTAGGAAAAGAGGCTACTTATGATGAAATCAAAGCTGCTATCAAGGCAGCGTCTGAGGGAAACATAGAAAGGGGAAATAACTTAAGGGTTAAACAAACCGAAAAGGAAAATAAAGAAAACGTAGGTATCCACTGACTTTGTGGGCGACAGCAGGTCGAGCATCTTTGTATTCCAAGGCTGGGATTGCTTTGTAATGATTCTTTTGTTAAGTTGGTGGCATGGTATGACAACAAGTGGAGGCCACACCTTCATGCACTGAACTCTTCCACCATTGCTGGCCAAAGAGGTGATATCAGTGTCAAGCTTCCGGGTTACCGGGAAAGCCGATGTGGACTTGAGGCCGGTGACGGAGCAACCATGCTGGCCTGAGCCGGAGCAACTACAGCAGATGAGATTATGGAGGCAGCCATATCTG

 As written in the first review, it is GAPDH (Not GADPH as wrongly written in line 179 and Table S1). It stands for glyceraldehyde-3-phosphate dehydrogenase.

 Please, mark primers in listed sequences  (supplementary data including the new CsAAAT1 and CsAAAT2 sequences)

Lines 13, 217, 461 and Table 3: holly oil fragrance contains 2-hydroxy benzoic acid methyl ester. Please, revise or cite the used name in the article;  2-hydroxy-benzoicacimethylester (if the authors see it is more common).

Author Response

I thank the authors for their effort to improve the article. Now I can also see the submitted supplementary data. I strongly recommend the authors to carefully consider these points as they still affect the article methodology, hence the main conclusion:

Q1: qPCR program must have a phase at its end to record the melt-curve, which is still missing. Please, add it to the program and provide the melt-curve as mentioned in the authors' response.

Response:Thank you for your suggestion. We apologize for our carelessness. We have added the program which is to record the melt-curve(Line 189-190).

“... , followed by 60 ℃ to 95 ℃ melting curve detection” 

We have also uploaded melt-curve plot(As shown in the following) to the supplementary materials as Figure S1.

What’s more, we have also added the procedures and parameters for high-temperature(95 ℃, 30s) denaturation and extension(72 ℃, 10 min). (Line 187-190).

Q2: Line 186: please add a statement that fresh tea leaves were used as control tissue where the relative expression was set to 2^0 (i.e. 1), also, in legend of figure 7.

Response:Thank you for your suggestion. I have added the instruction in Line 193-194 and the legend of Figure 7 (Line 313-314).

Q3: The answer of Q13 is still unclear to me. If re-alignment of the amplicons was performed, the experimentally obtained qPCR amplicons should have been sequenced. If this is the case, please add a clear statement that specificity of the amplicons was confirmed via sequencing.

Response: Thank you for your reminder. I have added add a statement that specificity of the amplicons was confirmed via sequencing.(Line 182-183)

Q4: Please, provide the 9 CsAAAT1 and 8 CsAAAT2 sequences as supplementary data or provide their accession numbers if you have submitted them to NCBI.

Response: Thank you for your suggestion. I have provided the 9 CsAAAT1 and 7 CsAAAT2(the number of CsAAAT2 is 7, not 8) nucleic acid sequence(5’-3’) as supplementary data(Line 259-260).

“The nucleic acid sequence of these two groups genes was presented in Table S3

Q5: Please, confirm that you have replaced figure 5 (and the upper display of Figure 6) with the newly constructed ML tree. It is uncommon to have 100% identical NJ and ML trees.

Response: Thank you for your question and reminder. We uploaded improved figures including figure 5 with ML tree to the system as ZIP file, but we forgot to replace the NJ tree in the manuscript. We were really sorry for our carelessness. And we have replaced the NJ tree with the Maximum Likelihood (ML) tree. (Line 280, Figure 5).

Q6: Line 179: The added accession number (KA295375.1) is for TSA. Fine, as the sequence still show 95% of a predicted CsGAPDH. I have copied NCBI sequence below because I cannot find the forward primer or the reverse complementary of the reverse primer shown in Table S1. This may affect the reliability of the chosen primers. 

>KA295375.1 TSA: Camellia sinensis tea_rep_c16164 mRNA sequence

GGCTCTGACAAAGAGACGATGTTGAGCTCGTCGCTGTCAACGACTCCCTTCATCACCACTGATTACATGACTTACATGTTCAAGTATGAAAAAAAAATGTGTTCACGGTCAATGGAAGCATCATGAACTCAAAGTGAAGGATTCCAAGACCCTTCTCTTTGGTGAGAAAGCAGTAGCTGTTTTTGGCCTTAGGAACCCAGAGGAGATCCCATGGGCTGAGACTGGAGCCGAATTCATTGTGGAGTCCACTGGTGTCTTCACTGATAAGGATAAGGCTGCTGCCCACTTGAAGGGTGGTGCAAAGAAGGTTATCATTTCAGCCCCTAGTAAGGATGCTCCCATGTTTGTTATGGGTGTCAATGAGAAGGATTACAAGCCAGATCTTCACATTGTTTCCAATGCTAGCTGCACTACCAACTGTCTTGCCCCCCTTGCTAAGGTTATCAATGACAAGTTTGTCATCGTTGAGGGTCTCATGACTACCGTGCACTCCATCACAGCCACACAAAAAACTGTTGATGGACCATCAAGCAAGGACTGGAGAGGTGGAAGAGCTGCTTCATTCAATATCATTCCTAGCAGCACTGGAGCTGCCAAGGCTGTTGGTAAGGTGCTGCCTGCACTTAATGGGAAATTGACTGGAATGGCTTTCCGCGTTCCCACTGTCGATGTCTCAGTGGTTGACCTCACTGTGAGGCTAGGAAAAGAGGCTACTTATGATGAAATCAAAGCTGCTATCAAGGCAGCGTCTGAGGGAAACATAGAAAGGGGAAATAACTTAAGGGTTAAACAAACCGAAAAGGAAAATAAAGAAAACGTAGGTATCCACTGACTTTGTGGGCGACAGCAGGTCGAGCATCTTTGTATTCCAAGGCTGGGATTGCTTTGTAATGATTCTTTTGTTAAGTTGGTGGCATGGTATGACAACAAGTGGAGGCCACACCTTCATGCACTGAACTCTTCCACCATTGCTGGCCAAAGAGGTGATATCAGTGTCAAGCTTCCGGGTTACCGGGAAAGCCGATGTGGACTTGAGGCCGGTGACGGAGCAACCATGCTGGCCTGAGCCGGAGCAACTACAGCAGATGAGATTATGGAGGCAGCCATATCTG

Response: Thank you for your reminder and thoughtfulness. After Sequence alignment (as shown in the screenshot below), we found that there was indeed an error in one base of each upstream and downstream primer(as shown in the following), which might be related to the old version of the internal reference gene used by our research group previously. We have corrected the primer list(Table S1). Thank you again for your suggestions.

Q7:  As written in the first review, it is GAPDH (Not GADPH as wrongly written in line 179 and Table S1). It stands for glyceraldehyde-3-phosphate dehydrogenase.

Response: Thank you for your reminder. We did write the abbreviated name of this reference gene incorrectly, and we are very sorry for this. We have corrected CsGADPH to CsGAPDH (Line183, Table S1).

Q8: Please, mark primers in listed sequences  (supplementary data including the new CsAAAT1 and CsAAAT2 sequences)

Response: Thank you for your question. We have marked the forward and reverse primers we designed in the CsAAAT1 and CsAAAT2 sequences in Table S2.

Q9: Lines 13, 217, 461 and Table 3: holly oil fragrance contains 2-hydroxy benzoic acid methyl ester. Please, revise or cite the used name in the article;  2-hydroxy-benzoicacimethylester (if the authors see it is more common).

Response: Thank you for your suggestion. Indeed, the “ 2-hydroxy benzoic acid methyl ester” is more common in papers. We agree with you, and we have replaced the 2-hydroxybenzoic acid methyl ester with the 2-hydroxybenzoic acid methyl ester (Lines 13, 225-226, 367-369, 473, and Table 3).

Reviewer 3 Report

I have no comments to the contents of the paper.

However, the manuscript needs a double check.

Figures still need attention. Please increase the text size to make it readable (see Instructions for Authors).

Minor editing of English language required.

Author Response

However, the manuscript needs a double check.

Figures still need attention. Please increase the text size to make it readable (see Instructions for Authors).

Response:Thank you for your encouragement and suggestion. We have carried out a double check of our manuscript, and we have improved all the figures and tables.

We have do our attempt to increase the text size to make it readable according to the Instructions for Authors of MDPI.

Thank you once again.

Round 3

Reviewer 2 Report

Thank you for your patience and careful revision. In my view, the article is a very acceptable state now, and I do not have more points to add.